# Predictive Control of Multi-Phase Motor for Constant Torque Applications

**Manuel R. Arahal [1,†]**[ID]**, Federico Barrero [2,*,†]**[ID]**, Manuel G. Satué [1]**[ID] **and Daniel R. Ramírez [1]**[ID]

1    Systems Engineering and Automation Department, University of Seville, 41092 Seville, Spain;
     arahal@us.es (M.R.A.); mgarrido16@us.es (M.G.S.); danirr@us.es (D.R.R.)
2    Electronics Engineering Department, University of Seville, 41092 Seville, Spain
*    Correspondence: fbarrero@us.es; Tel.: +34-954-487-372
†    These authors contributed equally to this work.

**Abstract:** Constant torque motors are needed for rotary screw compressors that are used for cooling and other applications. In such systems, the torque demanded by the load is approximately the same over the whole range of mechanical speeds. In this paper, the use of multi-phase induction machines is investigated for this type of application. The requirement of low stator current distortion is considered. A scheduled approach is used to provide the best possible tuning for each operating point, similar to the concept of gain scheduling control. Simulations and laboratory tests are used to assess the proposal and compare it with finite-state predictive control. The experiments show that a trade-off situation appears between the ripple content in stator currents in the torque-producing and harmonic planes. As a consequence, the controller tuning appears as an important step. The proposed method considers various figures of merit with cost function tuning, resulting in a scheduled scheme that provides improved results. It is shown that the approach leads to a reduction in current ripple, which is advantageous for this particular application.

**Keywords:** rotary screw compressors; multi-phase systems; predictive control; variable-speed drives

## 1. Introduction

Rotary screw air compressors (RSAC) have two meshed rotors that force the air to move into progressively smaller volumes, thus increasing the pressure. Variable-speed drives (VSD) are used in most cases for better energy efficiency [1]. However, the speed–torque curves for RSACs are very flat across the full speed range. This is very different from the torque curves of centrifugal pumps and compressors. Most recent cooling installations use RSACs driven by an adequate VSD capable of handling the constant torque profile [2].

Multi-phase induction machines (IM) have intrinsic characteristics that are useful for this type of application with respect to conventional three-phase systems. Among these, lower torque pulsation and DC-link current harmonics, higher overall system reliability, and better power distribution per phase are often mentioned [3]. These superior characteristics have made them a subject of research for novel applications in conjunction with new converters [4]. In addition, new configurations have been proposed with added degrees of freedom [5]. In the case of RSACs, these characteristics can be exploited to allow for smoother operation of the compressor, benefiting the whole installation. However, the multi-phase IM requires adequate control of the additional degrees of freedom introduced by the extra number of phases [6].

Model predictive control (MPC) has been applied to many types of systems including those with binary inputs [7]. This has favored its use as a versatile control technique for VSDs [8]. Many variants of MPC have been proposed in connection with electrical systems, and control strategies for tracking of current, torque, and speed have been successfully implemented [9]. In most cases, an enhancement over the standard approach is sought. For example, the torque ripple and the stator-current harmonic content can be mitigated using

non-fixed discretization times [10], redefining the utilized voltage vectors [11–13], using new cost functions [14], using identification algorithms [15], or modifying the number of active power legs [16,17]. Other factors such as switching frequency [18] and harmonic content [19] have also appeared in the literature.

In the realm of multi-phase drives, finite-state MPC (FSMPC) is arguably the most popular method for stator-current tracking. FSMPC easily allows the extra number of phases to be treated and provides high bandwidth since the voltage source inverter (VSI) is directly commanded by the FSMPC without the need for a modulation stage. Such a configuration, however, requires high computational power because the model-based control action computation must be performed in time periods of the order of microseconds [20]. The FSMPC implementation uses a cost function penalizing deviations from objectives [21]. The tuning of the objective function is not trivial, and FSMPC faces a trade-off between conflicting criteria [22,23]. Alternatively, some researchers have turned their attention to schemes avoiding weighting factors (WFs) [24]. In this paper, a methodology for tuning the predictive controller in constant torque applications is designed and analyzed. Because WFs bring flexibility that cannot be achieved otherwise, they are not eliminated but rather kept and selected considering multiple operating points, which allows a more effective tuning procedure [25]. A simple tuning procedure is provided here, where a scheduling variable is used in a way similar to in a previous study [23].

The next section presents the FSMPC scheme for the case study: a five-phase IM. Section 3 introduces the utilized assessment analysis. The proposal is then detailed in Section 4, including simulations and experimental results. Conclusions are drawn in the final section.

## 2. Case Study

The system under study was a five-phase IM fed by a two-level five-phase VSI (SKS 22F Semikron module). The IM was based on a three-phase IM with 30 slots, and was rewound in a starting connection configuration with an isolated neutral point, resulting in five phases and three pairs of poles. The system was regulated by a TMS320LF28335 digital signal processor (DSP). A digital encoder (GHM510296R/2500) and the enhanced quadrature encoder pulse peripheral of the DSP were used to measure the rotor mechanical speed, while the demanded load torque was set using a DC machine that was mechanically coupled to the five-phase drive. Photographs are shown in Figure 1. Note that the DC-link voltage was set to 300 V using an external DC power supply. This value and values for the other variables are given in Table 1.

**Table 1.** Five-phase IM parameters and electrical limits.

| Parameter | Value | Variable | Limit |
|---|---|---|---|
| Stator resistance, $R_s$ | 12.85 $\Omega$ | Voltage limit $V_{DC}$ | 300 V |
| Rotor resistance, $R_r$ | 4.80 $\Omega$ | Current limit, $I_{VSI}$ | 2.5 A |
| Stator leakage inductance, $L_{ls}$ | 79.93 mH | Rated d-current, $I_{sd,rated}$ | 0.9 A |
| Rotor leakage inductance, $L_{lr}$ | 79.93 mH | Maximum torque, $T_{em,max}$ | 8.13 Nm |
| Mutual inductance, $L_m$ | 681.7 mH | | |
| Rotational inertia, $J_m$ | 0.02 kg m$^2$ | | |
| Friction, $B_m$ | 0.0118 Nms/rad | | |
| Number of pairs of poles, $P$ | 3 | | |

An outer speed regulation loop based on a PI controller and indirect vector control was used as indicated in Figure 1. The FSMPC regulates the stator current using a discrete model of the IM to compute the state of the VSI for the next sampling period $u_j$ by minimization of the cost function $J$. An exhaustive search is used for the optimization, considering every available switching state ($2^5 = 32$ for the five-phase IM).

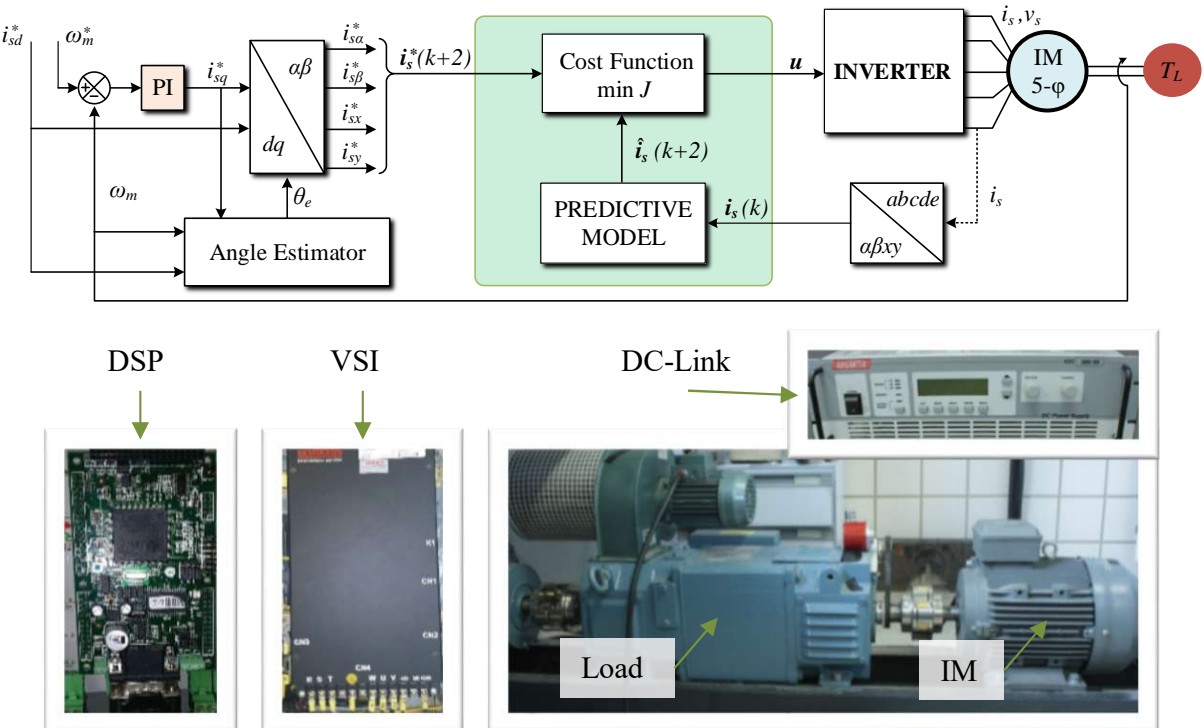

**Figure 1.** Diagram of FSMPC and photographs of the experimental test rig.

### 2.1. Five-Phase Induction Motor Model

For this particular case, and considering standard assumptions (uniform air gap, sinusoidal magnetomotive force distribution, and negligible core and magnetic losses), the following equation can be found (the reader is referred to [26] for a detailed description of the model and the reference frames).

$$\frac{dx}{dt}(t) = A_c(\omega_r(t))x(t) + B_c v(t) \tag{1}$$

The state vector is composed of $\alpha - \beta$ and $x - y$ stator currents and $\alpha - \beta$ rotor currents: $x = (i_{s\alpha}, i_{s\beta}, i_{sx}, i_{sy}, i_{r\alpha}, i_{r\beta})^T$. In addition, $v = (v_{s\alpha}, v_{s\beta}, v_{sx}, v_{sy})^T$ are the applied stator voltages. Matrices $A$ and $B$ depend on the rotor electric speed $\omega_r$.

$$A_c = \begin{bmatrix} -a_{s2} & a_{m4} & 0 & 0 & a_{r4} & a_{l4} \\ -a_{m4} & -a_{s2} & 0 & 0 & -a_{l4} & a_{r4} \\ 0 & 0 & -a_{s3} & 0 & 0 & 0 \\ 0 & 0 & 0 & -a_{s3} & 0 & 0 \\ a_{s4} & -a_{m5} & 0 & 0 & -a_{r5} & -a_{l5} \\ a_{m5} & a_{s4} & 0 & 0 & a_{l5} & -a_{r5} \end{bmatrix} \tag{2}$$

$$B_c = \begin{bmatrix} c_2 & 0 & 0 & 0 \\ 0 & c_2 & 0 & 0 \\ 0 & 0 & c_3 & 0 \\ 0 & 0 & 0 & c_3 \\ -c_4 & 0 & 0 & 0 \\ 0 & -c_4 & 0 & 0 \end{bmatrix} \tag{3}$$

The coefficients used are: $c_1 = L_s L_r - L_m^2$, $c_2 = L_r/c_1$, $c_3 = 1/L_{ls}$, $c_4 = L_m/c_1$, $c_5 = L_s c_1$, $a_{s2} = R_s c_2$, $a_{s3} = R_s c_3$, $a_{s4} = R_s c_4$, $a_{r4} = R_r c_4$, $a_{r5} = R_r c_5$, $a_{l4} = L_r c_4 \omega_r$, $a_{l5} = L_r c_5 \omega_r$, $a_{m4} = M c_4 \omega_r$, and $a_{m5} = M c_5 \omega_r$.

Therefore, the state of the VSI is utilized to compute the resulting stator voltages in the $\alpha - \beta$ and $x - y$ planes, $v = (v_{s\alpha}, v_{s\beta}, v_{sx}, v_{sy})$, $v = V_{DC}u T_M$, where $V_{DC}$ is the DC-link voltage, $u$ is a row vector containing the gating signals, and $T_M = T \cdot M$, with $T$ being the connectivity matrix of the VSI and $M$ being a coordinate transformation matrix. Their product is found as follows:

$$
T_M = \frac{2}{25} \begin{bmatrix} 4 & -1 & -1 & -1 & -1 \\ -1 & 4 & -1 & -1 & -1 \\ -1 & -1 & 4 & -1 & -1 \\ -1 & -1 & -1 & 4 & -1 \\ -1 & -1 & -1 & -1 & 4 \end{bmatrix} \cdot \begin{bmatrix} 1 & \cos\vartheta & \cos 2\vartheta & \cos 3\vartheta & \cos 4\vartheta \\ 0 & \sin\vartheta & \sin 2\vartheta & \sin 3\vartheta & \sin 4\vartheta \\ 1 & \cos 2\vartheta & \cos 4\vartheta & \cos\vartheta & \cos 3\vartheta \\ 0 & \sin 2\vartheta & \sin 4\vartheta & \sin\vartheta & \sin 3\vartheta \\ 1/2 & 1/2 & 1/2 & 1/2 & 1/2 \end{bmatrix} \quad (4)
$$

where $\vartheta = 2\pi/5$. Equation (1) is discretized with the sampling time $T_s$ to be used in the predictive model. A two-step-ahead prediction is needed to account for the fact that the computation time takes most of the sampling time [8]. The following expression can be found for the prediction $\hat{\imath}(k+2|k)$ for stator currents:

$$
\hat{\imath}(k+2|k) = Ai(k) + B_1 u(k) + B_2 u(k+1) + \hat{G}(k|k) \quad (5)
$$

Note that $u(k)$ and $u(k+1)$ are the actual and next control actions, and $\hat{G}(k|k)$ is a term accounting for the dynamics of the rotor currents, which are not usually measured for practical and economical reasons.

### 2.2. Constant Torque Applications

As previously described, the controller uses an outer feedback loop for torque/speed regulation, using an indirect vector control technique and managed by a PI controller. The electric torque developed by the IM must overcome the opposing torque $T_L$, friction, and inertia. Then, $T_e = T_L + B_m \omega_m + J_m \dot{\omega}_m$, where $B_m$ is the friction parameter and $J_m$ is the inertia (see Table 1 for a list of IM parameters, their actual values, and electrical limits). To produce the appropriate electrical torque, the $d - q$ components of the stator currents must be set adequately. The flux component $i_{sd}^*$ is set by the nominal flux ($\phi_n$), so that $i_{sd}^* = \phi_n/L_m$. The quadrature component $i_{sq}^*$ is provided by the PI controller. As a result, in the steady state, the following expressions are verified:

$$
T_e = T_L + B_m \omega_m = P \frac{5}{2} \frac{L_m^2}{L_r} i_{sd}^* i_{sq}^*, \quad (6)
$$

and it is possible to compute the quadrature component needed to sustain a certain speed as

$$
i_{sq}^* = \frac{T_L + B_m \omega_m}{i_{sd}^*} \frac{L_r}{PL_m^2} \frac{2}{5}. \quad (7)
$$

The electrical frequency can be found by considering the mechanical rotor speed and the slip frequency from $\omega_e = \omega_r + \omega_{sl}$, where the field orientation is assumed and $\omega_{sl} = \frac{R_r}{L_r} \frac{i_{sq}^*}{i_{sd}^*}$. This allows us to write the following relationship for this particular case:

$$
\omega_e = P\omega_m + \frac{T_L + B_m \omega_m}{\left(i_{sd}^*\right)^2} \frac{R_r}{PL_m^2} \frac{2}{5} \quad (8)
$$

The inner and outer loops are then connected due to the indirect field-oriented control scheme, where the flux and the electrical torque are independently controlled using reference currents $i_{sd}^*$ to regulate the flux and $i_{sq}^*$ to control the electrical torque. In this configuration, the reference currents in the $d - q$ plane are translated to the $\alpha - \beta$ space

using the Park transformation, obtaining a reference for the stator current in the $\alpha - \beta$ plane, $I^*_{\alpha-\beta} = D\left(i^*_{sd}i^*_{sq}\right)^{\mathsf{T}}$, where the matrix $D$ is given by

$$D = \begin{bmatrix} \cos \theta_a & \sin \theta_a \\ -\sin \theta_a & \cos \theta_a \end{bmatrix} \tag{9}$$

and the flux position $\theta_a$ is obtained as $\theta_a = \int \omega_e dt$. The reference signal for the stator current tracking $i^*(k)$ uses an amplitude $I^*_s$ computed as $I^*_s = \sqrt{i^{*2}_{sd} + i^{*2}_{sq}}$, which is found to be a function of the mechanical speed by virtue of the previous expressions in Equations (7) and (8). As a result, the references for stator currents are $i^*_{s\alpha}(t) = I^*_s \sin \omega_e t$, $i^*_{s\beta}(t) = I^*_s \cos \omega_e t$, $i^*_{sx}(t) = 0$, and $i^*_{sy}(t) = 0$.

The selection of $u(k+1)$ at the discrete time $k$ is made by minimizing the objective function for time $k+2$ and $J(k+2)$, because the computations needed by the FSMPC take up most of the sampling period. Hence, the computed control signal is released at $(k+1)$ and affects the output at $(k+2)$ [8]. This objective function can incorporate a number of different terms to convert a multi-objective optimization problem into a one-objective problem [27]. The simplest objective function penalizes the predicted control error $\hat{e}(k+2) = \left(i^*(k+2) - \hat{i}(k+2)\right)$, where $i^*(k+2)$ represents the reference for the state space vector $i$, and $\hat{i}(k+2)$ represents the two-step-ahead prediction. More complex objective functions are used to regulate $x - y$ currents. With these considerations, the cost function to be applied can be written as

$$J(k+2) = \|\hat{e}_{\alpha\beta}(k+2)\|^2 + \lambda_{xy}\|\hat{e}_{xy}(k+2)\|^2 \tag{10}$$

where $\|.\|$ denotes the vector modulus, $\hat{e}_{\alpha\beta}(k+2) = i^*_{s\alpha\beta}(k+2) - \hat{i}_{s\alpha\beta}(k+2)$ is the predicted tracking error in the $\alpha - \beta$ plane, $\hat{e}_{xy}(k+2) = \hat{i}_{sxy}(k+2)$ is the predicted tracking error in the $x - y$ plane, and $\lambda_{xy}$ is the WF for this cost function.

## 3. Figures of Merit and Preliminary Analysis

The usual practice in MPC for drives is to tune the controller by selecting a cost function with a given structure and WFs. The tuning goal is to achieve the particular compromise solution that is deemed best on a global basis [23,28]. Several figures of merit have been used for VSDs, such as torque ripple, speed ripple, average switching frequency (ASF), total harmonic distortion (THD), stator current error, and $x - y$ current content. These figures of merit are not independent as there are some links due to the IM dynamics and the closed-loop operation. For instance, the tracking error of stator currents in $\alpha - \beta$ subspace is directly related to the current THD since the reference current is sinusoidal. Speed ripple is a low-pass filtered version of the torque ripple due to mechanical inertia. Torque ripple directly depends on the ripple of the $\alpha - \beta$ currents. Tracking in the $x - y$ plane affects the stator copper losses and THD. Consequently, keeping the $x - y$ currents close to zero simultaneously improves the power quality and drive efficiency. The switching frequency is not constant, and its values are important for the selection of the hardware (e.g., standard IGBTs or SiC-based power switches) and the efficiency (VSI losses). An average value, referred to as the average switching frequency (ASF), is usually considered in FSMPC. With this in mind, the following efficiency factors are considered:

$$E_{\alpha-\beta} \quad = \quad \sqrt{\frac{1}{(k_2 - k_1 + 1)} \sum_{k=k_1}^{k_2} e_{\alpha\beta}^2(k)} \tag{11}$$

$$E_{x-y} \quad = \quad \sqrt{\frac{1}{(k_2 - k_1 + 1)} \sum_{k=k_1}^{k_2} e_{xy}^2(k)} \tag{12}$$

$$ASF \quad = \quad \frac{1/5}{T_s(k_2 - k_1 + 1)} \sum_{k=k_1}^{k_2} \Delta S(k) \tag{13}$$

$$THD \quad = \quad = \frac{100}{I_1} \sqrt{\sum_{i=2}^{\infty} I_i^2} \tag{14}$$

where $\Delta S(k) = \sum_{i=1}^{5} |u_i(k+1) - u_i(k)|$ is the number of switch changes produced at the VSI when configuration $u(k)$ is changed to $u(k+1)$, and $I_i$ is the amplitude of the *i*-th harmonic component of the stator current. These quantities are defined over a temporal horizon defined by the discrete-time indices $k_1$, $k_2$.

Some preliminary simulations are shown to illustrate the problem of tuning the MPC for this particular application. The IM was simulated using the classical fourth-order Runge–Kutta algorithm implemented using the ode4 MATLAB function (2014 version). In addition, the controller was considered as a discrete-time subsystem with a sampling time of 66.6 ($\mu$s), equivalent to a 15 (kHz) sampling rate. This value is within the range usually found in papers dealing with predictive stator current control. In addition, this value can be obtained by a variety of digital signal processors, including the one utilized in this case (TMS320LF28335). The outer loop was simulated to obtain a steady state for each considered speed and the corresponding load torque and friction torque.

To provide a complete picture of the drive behavior, a set of values for $(\lambda_{xy}, \omega_m)$ were chosen in a lattice. For each $(\lambda_{xy}, \omega_m)$ combination, the simulation was run for a time sufficient to include a dozen electrical cycles at frequency $f_e$, so that $k_2$ was chosen as $k_2 = k_1 + N_c f e T_s$. After each simulation, the figures of merit were computed and shown as maps (see Figure 2), for the figures of merit defined in (11)–(14). It can be seen that the figures of merit are linked to one another. Furthermore, it is worth noting that the relationships are nonlinear as the surfaces have changing curvature and are not even convex. This makes the tuning of the controller a difficult task. Several aspects deserve comment:

- Both $E_{\alpha-\beta}$ and $E_{x-y}$ have a mostly monotonic variation with $\omega_m$ and $\lambda_{xy}$. However, if $\lambda_{xy}$ is chosen to achieve a low $E_{\alpha-\beta}$, then the corresponding $E_{x-y}$ is larger. This effect, however, is more marked at some speeds than others.
- The ASF map shows a bump around mid-range, with lesser values found for extreme values of $\lambda_{xy}$. This constitutes a major complication for standard tuning procedures. Nevertheless, the maximum ASF for all tunings and speeds is acceptable for most applications except, perhaps, very high-power applications.
- The THD depends mostly on speed and very little on $\lambda_{xy}$, and this is the reason for using an edge-on presentation in the last plot.

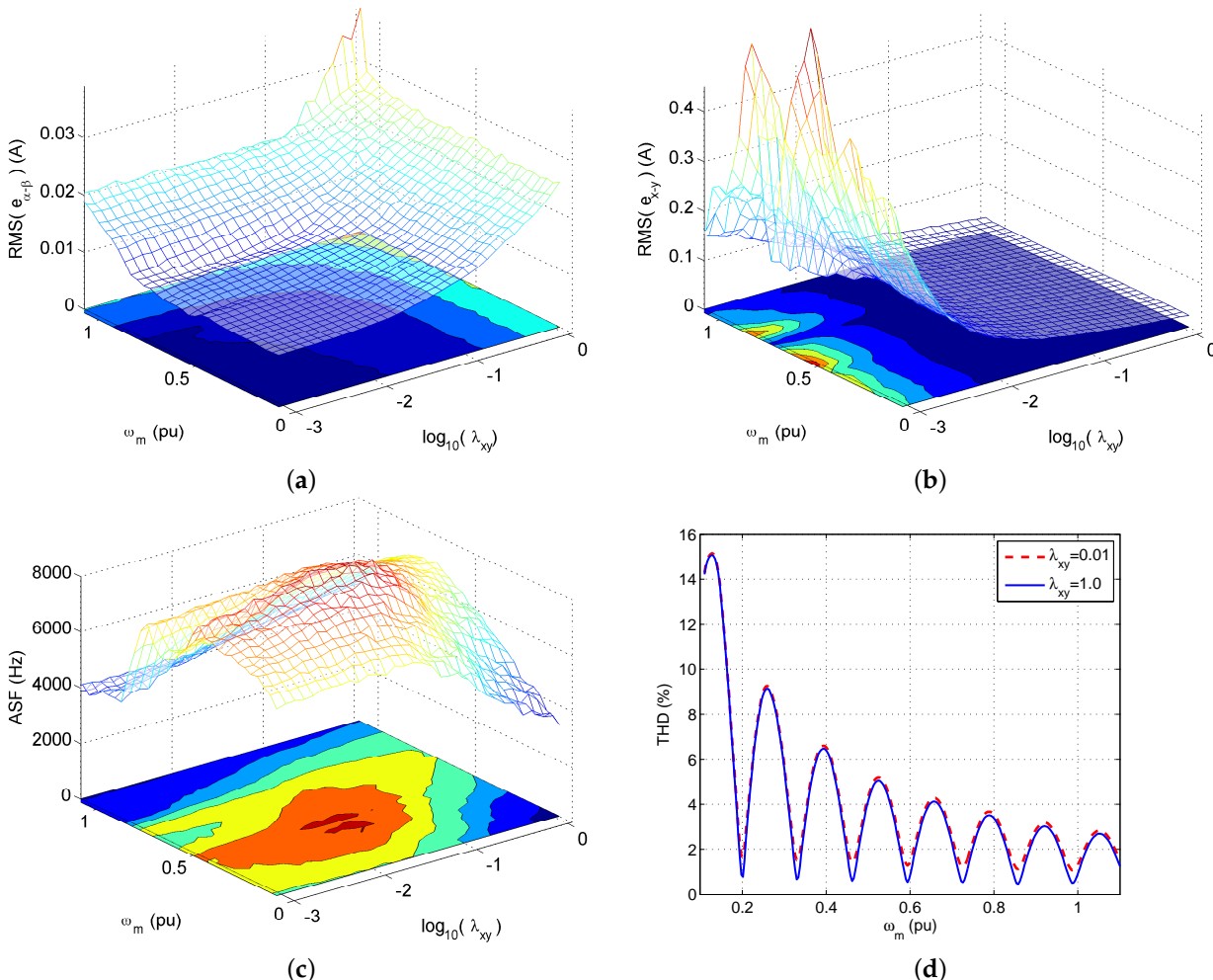

**Figure 2.** Performance map for the figure of merit over the whole speed range ($\omega_m$) and for all controller tunings ($\lambda_{xy}$), defined in Equation (11) (**a**), in Equation (12) (**b**), and in Equation (13) (**c**). (**d**) Performance plots for the figure of merit defined in eq. 14 over the speed range, using ($\lambda_{xy} = 1$ (solid blue line) and ($\lambda_{xy} = 0.01$ (broken red line) for the controller tuning.

## 4. Proposed Tuning and Assessment

Some objectives and limitations are considered in the tuning procedure. It will be assumed that the VSI imposes a limit on $F_{sw}$. Then, the WF tuning should provide a value of $F_{sw}$ below the limit for all possible operating points. This condition can be expressed as $F_{sw} < U_{sw}$. In terms of performance, the current ripple should be minimized to provide the load with a smooth velocity and prevent the possible appearance of oscillations. In addition, the $x - y$ currents are a source of inefficiency as they do not produce torque, only copper losses, and thus they should be minimized.

Two tuning strategies will be compared: (1) the standard strategy where WFs are fixed, with the resulting controller referred to as Std-FSMPC, and (2) a scheduled strategy, where $\omega_m$ is taken as the scheduling variable and a different WF is set for each $\omega_m$ using a function or table. The resulting controller in this case is referred to as Sch-FSMPC. The tuning is derived from the performance maps presented in the previous section. From Figure 2, it is clear that: (1) THD does not depend significantly on tuning and (2) tuning based on $E_{\alpha-\beta}$ and $E_{x-y}$ must be based on a trade-off. The scheduling of $\lambda_{xy}$ can be obtained by considering the trade-off between $E_{\alpha-\beta}$ and $E_{x-y}$ separately for each speed. For this particular application, at lower speeds the required stator current has a lower amplitude.

Thus, a certain value of $E_{\alpha-\beta}$ is more noticeable at lower speeds, causing a more detrimental effect on speed control. With these considerations, the optimal WF is found to be

$$\lambda_{xy}^*(\omega_m) = \underset{\lambda_{xy}}{\operatorname{argmin}} H(\lambda_{xy}, \omega_m) \tag{15}$$

where $H$ is a function that penalizes values of $E_{\alpha-\beta}$ over the acceptable threshold for each $\omega_m$ and also penalizes values of $\lambda_{xy} > 0$. It can easily be constructed as

$$H = K\big(E_{\alpha-\beta} < U_{\alpha-\beta}(\omega)\big) + \lambda_{xy} \tag{16}$$

where $K$ is any number large enough such that minimizing $H$ always favours the fulfilment of the condition $E_{\alpha-\beta} < U_{\alpha-\beta}(\omega)$. For instance, $K = 10$ is a valid choice since in most applications $\lambda_{xy} \ll 10$.

Figure 3 shows the values of $\lambda_{xy}^*$ obtained for each speed. It is interesting to note that, in different papers dealing with FSMPC, constant values of $\lambda_{xy}$ in the interval $[0.1, 0.5]$ are usually found. The optimal values found in this case fall almost exactly within the mentioned range.

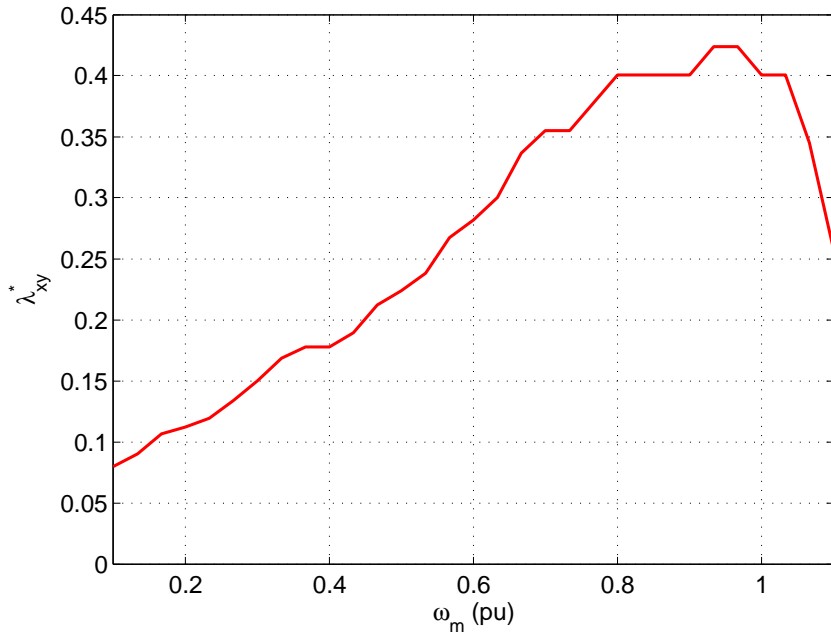

**Figure 3.** Optimal values of $\lambda_{xy}$ as a function of $\omega_m$.

### 4.1. Simulation Results

The proposed tuning scheme was compared with constant tuning via simulation. The same environment as in the preliminary simulations was used. The scheduled controller Sch-FSMPC used the WF values shown in Figure 3. The results are presented in Figure 4. It can be seen that the values of $E_{\alpha-\beta}$ and $E_{x-y}$ are linked by a trade-off so that when one decreases the other increases. However, thanks to the WF tuning provided by Equation (15), the $E_{x-y}$ values are reduced with respect to the standard case, while the performance is maintained in terms of $E_{\alpha-\beta}$.

Regarding the other figures of merit, first, the ASF is well below 6kHz, which is an acceptable value for most applications except perhaps for high-power ones. Second, as has been previously pointed out, the THD does not depend strongly on the tuning, so the values presented in Figure 2 are also valid here.

Continuing with the comparison, the following two tunings for the Std-FSMPC case are presented and compared with the proposal: the first (Std-FSMPC-a) corresponds to $\lambda_{xy} = 0.205$, and the second (Std-FSMPC-b) corresponds to $\lambda_{xy} = 0.410$. The comparison was carried out globally, that is, considering the whole speed range. Minimum and maximum values are reported for each tuning.

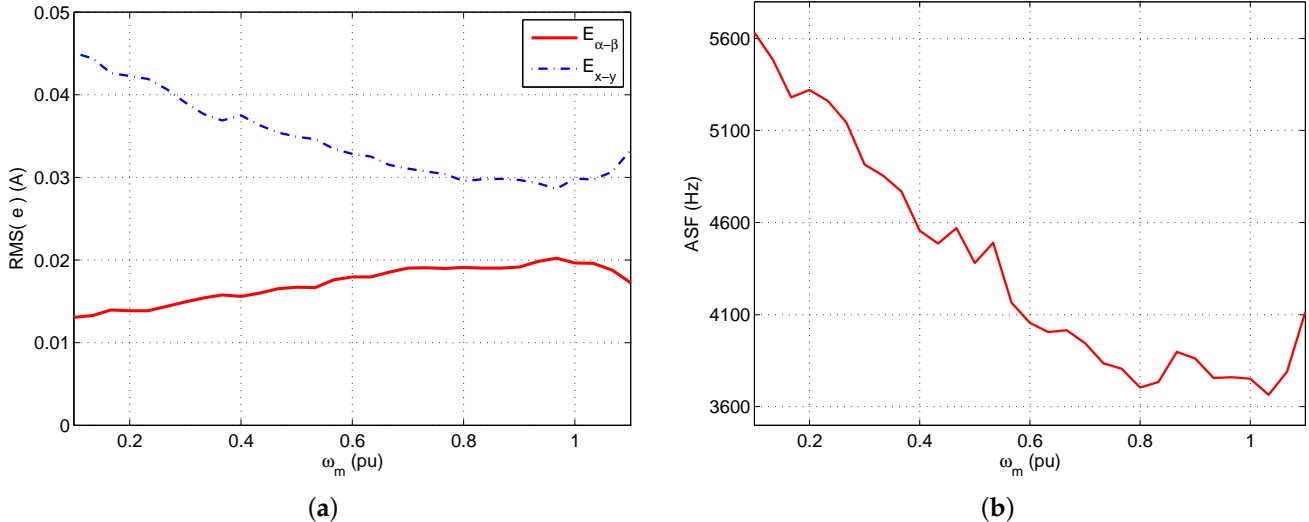

(**a**)          (**b**)

**Figure 4.** Simulation results for the proposed scheduled MPC over the whole range of speed of the IM: (**a**) $E_{\alpha-\beta}$ and $E_{x-y}$; (**b**) ASF.

It can be seen in Table 2 that the scheduled approach provides better results as it is able to reduce ripple without degrading the $x - y$ content and other factors. For instance, the first tuning (Std-FSMPC-a) produces an average value for $E_{\alpha-\beta}$ that is nearly the same as that found in the scheduled solution but with a higher $x - y$ content. The second tuning (Std-FSMPC-b) produces, in the worst case (highest speed), a value for $E_{\alpha-\beta}$ that is nearly the same as that found in the worst case of the scheduled solution but, again, with worse $x - y$ regulation.

**Table 2.** Summary of simulation results considering all operating points in the speed range. (Std = standard, Sch = scheduled).

| Controller | $E_{\alpha-\beta}$ min (A) | $E_{\alpha-\beta}$ max (A) | $E_{x-y}$ min (A) | $E_{x-y}$ max (A) | $F_{sw}$ min (Hz) | $F_{sw}$ max (Hz) |
|---|---|---|---|---|---|---|
| Std-FSMPC-a | 0.01591 | 0.01670 | 0.03493 | 0.05922 | 4075 | 4760 |
| Std-FSMPC-b | 0.01905 | 0.02005 | 0.02893 | 0.05569 | 3440 | 4053 |
| Sch-FSMPC | 0.01256 | 0.02005 | 0.02859 | 0.04630 | 3665 | 5751 |

Finally, a comparison is performed by simulation considering three operating regimes characterized by mechanical speeds of 150 rpm, 280 rpm, and 500 rpm, respectively. These regimes are also considered in the experimental tests. Table 3 shows the figures of merit for these regimes and for the two controllers. As before, (Std-) stands for the standard controller and (Sch-) for the proposed scheduled controller. It can be seen that the proposal does improve the current ripple while maintaining the other factors. It is worth commenting that the trade-offs between the figures of merit previously reported in [25] are evident in this case also. Moreover, and in line with previous results, the standard MPC tuning showed non-optimal behavior, and better results were found with the scheduled approach.

**Table 3.** Additional simulation results for particular operating points. (Std = standard, Sch = scheduled).

| Case | $\omega_m^*$ (rpm) | $i_{sq}^*$ (A) | Contr | $\lambda_{xy}$ | $E_{\alpha-\beta}$ (A) | $E_{x-y}$ (A) | $F_{sw}$ (Hz) | THD (%) |
|------|------|------|-----------|------|--------|-------|------|-----|
| A | 150 | 1.6 | Std-FSMPC | 0.20 | 0.0154 | 0.038 | 6096 | 8.1 |
| A | 150 | 1.6 | Sch-FSMPC | 0.30 | 0.0156 | 0.034 | 5111 | 8.0 |
| B | 280 | 1.8 | Std-FSMPC | 0.20 | 0.0162 | 0.037 | 6651 | 7.5 |
| B | 280 | 1.8 | Sch-FSMPC | 0.35 | 0.0164 | 0.031 | 5721 | 7.5 |
| C | 500 | 2.4 | Std-FSMPC | 0.20 | 0.0171 | 0.036 | 7668 | 7.4 |
| C | 500 | 2.4 | Sch-FSMPC | 0.45 | 0.0172 | 0.029 | 7111 | 7.1 |

*4.2. Experimental Results*

Various experimental tests were performed on the test bench depicted in Figure 1. The electrical and mechanical parameters of the five-phase machine used in the predictive model are detailed in Table 1. They were experimentally obtained in [29], where identification procedures similar to those described in [30] were applied.

Several operating points were considered for the assessment of the proposed scheme. The results of the proposed method and a standard FSMPC technique are shown in Table 4. Superior behavior was observed using the proposed method, verified for $\alpha - \beta$ tracking, for $x - y$ regulation, and with similar use of VSI commutations well below standard values such as 10 KHz. Comparing values in Tables 3 and 4, it is interesting to see that the simulation results provide lower values for most figures of merit. This is to be expected as there are no measurement errors or other experimental factors in the simulations. In addition, the experiments confirm that the values of THD depend mostly on speed and very little on $\lambda_{xy}$. Again, due to experimental factors, this finding is somewhat smeared out compared with the simulation results.

**Table 4.** Summary of results for comparison. (Std = standard, Sch = scheduled).

| Case | $\omega_m^*$ (rpm) | $i_{sq}^*$ (A) | Contr | $\lambda_{xy}$ | $E_{\alpha-\beta}$ (A) | $E_{x-y}$ (A) | $F_{sw}$ (Hz) | THD (%) |
|------|------|------|-----------|------|-------|-------|------|------|
| A | 150 | 1.6 | Std-FSMPC | 0.20 | 0.121 | 0.112 | 7153 | 8.95 |
| A | 150 | 1.6 | Sch-FSMPC | 0.30 | 0.121 | 0.105 | 7185 | 7.70 |
| B | 280 | 1.8 | Std-FSMPC | 0.20 | 0.130 | 0.115 | 6671 | 7.76 |
| B | 280 | 1.8 | Sch-FSMPC | 0.35 | 0.124 | 0.103 | 6628 | 6.18 |
| C | 500 | 2.4 | Std-FSMPC | 0.20 | 0.132 | 0.109 | 4469 | 7.61 |
| C | 500 | 2.4 | Sch-FSMPC | 0.45 | 0.125 | 0.104 | 6811 | 6.36 |

Figure 5 shows the trajectories of $i_s$ for the $\alpha$ and $x$ axes (similar results are found for $\beta$ and $y$ axes) for cases A to C. It can be observed that the stator current is regulated using both techniques to follow the reference values. However, the proposed scheme achieves less current ripple on both axes. This is more evident at low speed (cases a1 and a2), where the current controller must provide a lower value for the fundamental component (low modulation number), resulting in an increased THD. Recall that as per Equation (16), the tuning objective is to provide a lower harmonic distortion while maintaining the tracking ability of the standard FSMPC, measured by $E_{\alpha-\beta}$.

In addition to the previous tests, the transient response is considered in the following. Figure 6 shows the results for a speed step test. It can be seen that the speed reference is changed from 0 (rpm) to 500 (rpm), which is a large step. The rising time is about 1 s, which testifies to a bandwidth more than enough for the intended application as a motor for RSACs. In addition, the observed variations in speed are small and are mainly attributable to measurement noise.

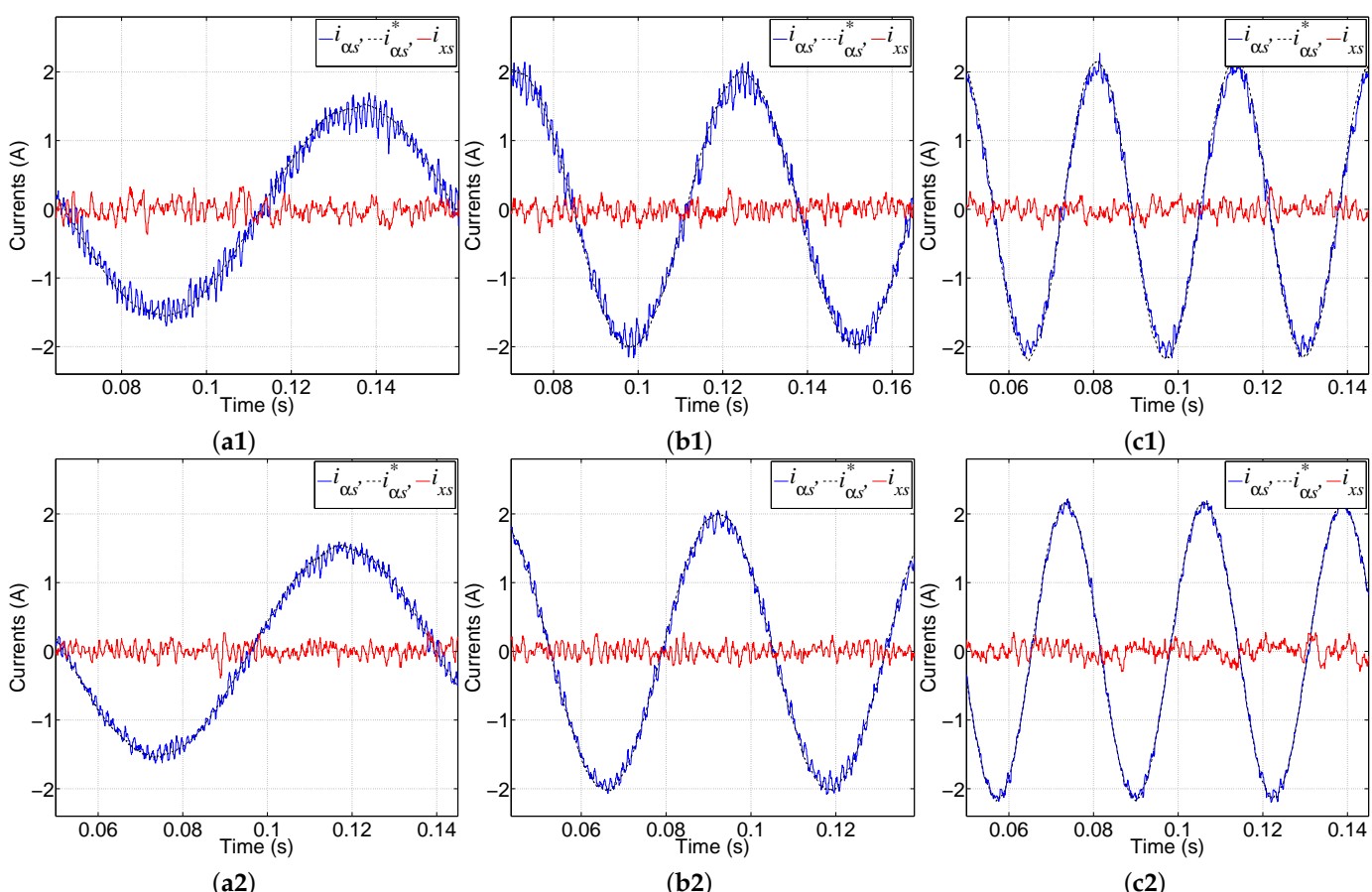

**Figure 5.** Experimental results for: (**a1**) case A with standard MPC tuning; (**a2**) case A with proposal; (**b1**) case B with standard MPC tuning; (**b2**) case B with proposal; (**c1**) case C with standard MPC tuning; (**c2**) case C with proposal.

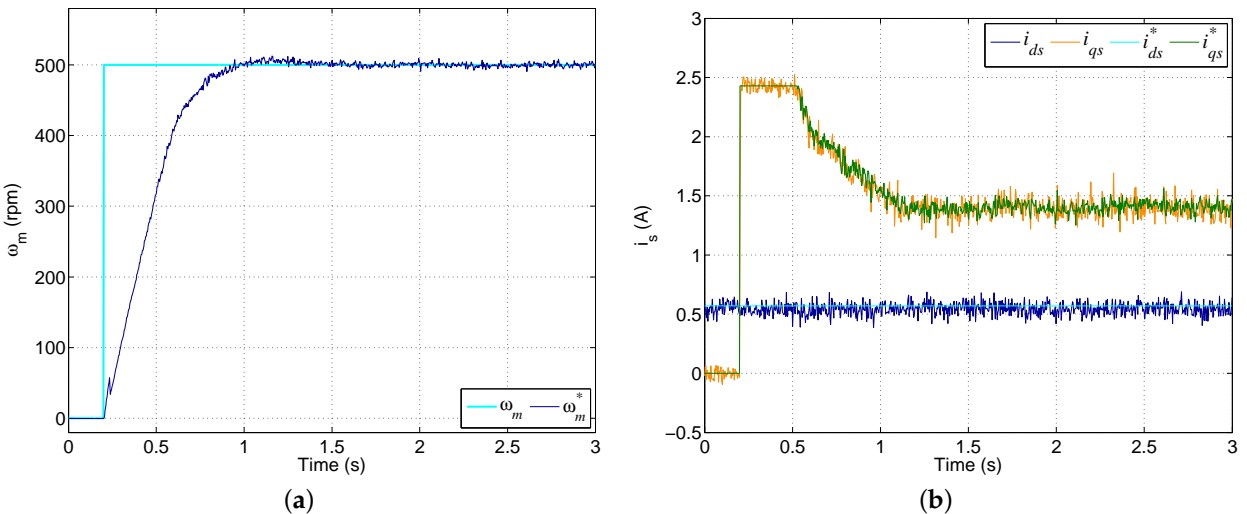

**Figure 6.** Experimental results for a step test showing the evolution of: (**a**) mechanical speed $\omega_m$; (**b**) torque-producing stator current $i_{sq}$ and flux component of stator current $i_{sd}$. For each variable, the reference value is also shown.

## 5. Conclusions

The conducted analysis showed interesting features of the use of the multi-phase IM for variable-speed and constant-torque applications.

It was shown that both $E_{\alpha-\beta}$ and $E_{x-y}$ have a mostly monotonic variation with $\omega_m$ and $\lambda_{xy}$. This produces a trade-off situation between the ripple content in the stator currents in both planes. The search for an appropriate value for $\lambda_{xy}$ is thus an important problem.

Besides ripple, other important aspects are the average switching frequency and the total harmonic distortion. It was shown that the ASF was below acceptable limits for all tunings and speeds. Moreover, the THD depended mostly on speed and very little on $\lambda_{xy}$. This phenomenon has not been reported before and has consequences for the tuning methodology. In particular, it allows for a simple tuning procedure for use with the scheduled approach taken in this paper.

It is also remarkable that the trade-offs between figures of merit previously reported in [25], also appeared in this case. Finally, it can be stated that the standard MPC tuning did not provide an optimal solution, whereas the proposed scheduled approach provided enhanced results that were advantageous for this particular application.

**Author Contributions:** Conceptualization and methodology, M.R.A. and F.B.; validation, M.R.A., F.B., and M.G.S.; formal analysis and investigation, M.R.A., M.G.S. and D.R.R.; resources and data curation M.R.A.; writing—original draft preparation, writing—review and editing, and visualization, M.R.A., F.B. and M.G.S.; supervision, project administration, and funding acquisition, M.R.A. and D.R.R. All authors have read and agreed to the published version of the manuscript.

**Funding:** This research was funded as Proyecto RTI2018-101897-B-I00 by FEDER/Ministerio de Ciencia e Innovación—Agencia Estatal de Investigación, and as grant P20_00546 by Junta de Andalucía and FEDER funds.

**Institutional Review Board Statement:** Not applicable.

**Informed Consent Statement:** Not applicable.

**Data Availability Statement:** Not applicable.

**Conflicts of Interest:** The authors declare no conflict of interest. The funders had no role in the design of the study; in the collection, analyses, or interpretation of data; in the writing of the manuscript, or in the decision to publish the results.

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
