# Peer review of "Predictive Control of Multi-Phase Motor for Constant Torque Applications"

_machines, doi:10.3390/machines10030211_

Round 1
Reviewer 1 Report
1) It is difficult for the reader to clearly see the control board, 2-level VSI, and the 5-phase IM and its programmable load as presented in Fig.1. The test rig photo that depicts the 5-phase IM and its load may be shown in a separate figure.
2) Some parameters in equation (8) are not defined.
3) Would you briefly elaborate how the electrical parameters were obtained? The reader would like to assert the accuracy of these parameters.
4) In addition to the DC link voltage, would you please provide the ratings of the 5-phase IM.
5) Fig. 5 may be improved to a better resolution.
6) A six-pole, 5-phase IM with 30 stator slots has one slot per pole per phase. Would you briefly tell the reader the type of winding configuration used. The reader would like to assert the contribution of your proposed method, and attribute less attention to the winding configuration. The later plays also a role.
7) The reader would like to see the analysis of the shaft torque and its ripple contents (Experimental results), in order to connect with the low stator current distortion.
8) Some simulation results may be compared with experimental results.
9) The abstract may be improved by describing the approach used and providing key findings.
10) More analysis of results is required, especially results presented under sections 4.1 and 4.2.
11) The conclusion is insufficient. A more elaborative conclusion is required, and possible recommendations to be included.
Author Response
The authors thank the anonymous reviewers for their work. We feel that, by incorporating their suggestions, the paper has improved in terms of quality and clarity. In what follows the author's responses to the reviewer’ comments are given on a point-to-point basis.
1) It is difficult for the reader to clearly see the control board, 2-level VSI, and the 5-phase IM and its programmable load as presented in Fig.1. The test rig photo that depicts the 5-phase IM and its load may be shown in a separate figure.
Please note that the figure has been modified in the revised version to solve this issue.
2) Some parameters in equation (8) are not defined.
The text accompanying previous equation (8), now equation (6), has been rewritten to solve the problem, including a reference to the table of parameters.
3) Would you briefly elaborate how the electrical parameters were obtained? The reader would like to assert the accuracy of these parameters.
Some new references have been included to illustrate the applied process for obtaining the electrical and mechanical parameters of our five-phase IM.
4) In addition to the DC link voltage, would you please provide the ratings of the 5-phase IM.
Solved in the revised document (please see Table 1).
5) Fig. 5 may be improved to a better resolution.
Figure 5 has been re-uploaded in vector format with large resolution. Please notice that the lines in the plot have width, so they can be properly rendered for visualization on paper or screen. This means that, after zooming in, they will become larger and wider. Nevertheless, it is possible to apply large zoom scales before this effect is noticeable.
6) A six-pole, 5-phase IM with 30 stator slots has one slot per pole per phase. Would you briefly tell the reader the type of winding configuration used. The reader would like to assert the contribution of your proposed method, and attribute less attention to the winding configuration. The later plays also a role.
The text has been updated to include details about the winding configuration for future readers.
7) The reader would like to see the analysis of the shaft torque and its ripple contents (Experimental results), in order to connect with the low stator current distortion.
New experimental results have been added to check the effect of the controller on mechanical variables.
8) Some simulation results may be compared with experimental results.
New simulation results have been added (Table 3) to be compared with experimental results (now Table 4).
9) The abstract may be improved by describing the approach used and providing key findings.
The abstract has been rewritten to include the approach used and provide key findings.
10) More analysis of results is required, especially results presented under sections 4.1 and 4.2.
The simulation and experimental results are commented in more detail in the revised version of the paper. Also, results are linked to WF tuning as per equations (15)-(16) that are the basis of the method.
11) The conclusion is insufficient. A more elaborative conclusion is required, and possible recommendations to be included.
The last section of the manuscript has been rewritten to clarify the obtained conclusions.

Reviewer 2 Report
The proposed paper presents a methodology, simulation, and experimental results on predictive control tuning of five-phase IM for constant torque applications.
The paper structure is clear and the paper itself could be of interest to the readers. However, I have some concerns that need to be addressed to improve the manuscript readability:
- please, justify the use of the two-step prediction horizon of MPC regarding the accuracy of the cost function;
- explicitly specify the benefits of the proposed approach in the abstract;
- specify the order of the Runge-Kutta method (line 203);
- add alphabetic designations for subfigures (Figures 2, 4, and 5);
- explicitly introduce the designations for the internal paragraph formula (lines 144-145);
- fix some minor typos (e.g. line 4).
Author Response
The proposed paper presents a methodology, simulation, and experimental results on predictive control tuning of five-phase IM for constant torque applications.
The paper structure is clear and the paper itself could be of interest to the readers.
The authors thank this reviewer for the appreciation of our research.
However, I have some concerns that need to be addressed to improve the manuscript readability:
- please, justify the use of the two-step prediction horizon of MPC regarding the accuracy of the cost function;
Please notice that the two-step ahead prediction is needed by FSMPC approaches, as reported first in [R1] and later on in a number of publications dealing with the FSMPC method.
[R1] Arahal, M. R., et al. "Multi-phase current control using finite-state model-predictive control." Control Engineering Practice 17.5 (2009): 579-587.
The revised document has been modified (see section 2.2) to clarify this issue and direct future readers to appropriate bibliography in the field.
- explicitly specify the benefits of the proposed approach in the abstract;
The abstract has been rewritten to include the approach used and provide key findings.
- specify the order of the Runge-Kutta method (line 203);
Some sentences have been added to Section 3 to clarify the way simulations are carried out, including the order of the RK method.
- add alphabetic designations for subfigures (Figures 2, 4, and 5);
The subfigures in Figures 2, 4, and 5 now have an alphabetic designation to help in their identification.
- explicitly introduce the designations for the internal paragraph formula (lines 144-145);
The text accompanying equation (8) has been rewritten so that all parameters are defined prior to their use. Also, a reference to Table 1 has been added to direct readers to further information about the parameters.
- fix some minor typos (e.g. line 4).
The entire document has been revised for English grammar and typos.

Reviewer 3 Report
- In the introduction part more literature review is needed.
- Could you explain what is the speed and positioning of x-y coordinate system?
- The matrices are usually put in square brackets.
- All abbreviations should be explained (e.g. ASF, HCSMPC).
- Where are two tunings of the STD-FSMPC presented (mentioned in rows 274-280)?
- Even though this article brings interesting simulation and experimental results that support the idea of better smoothing of current, and thus torque ripples in five-phase IM achieved with Sch-FSMPC, the results should be further analyzed and discussed in section 5.
Author Response
The authors thank the anonymous reviewers for their work. We feel that, by incorporating their suggestions, the paper has improved in terms of quality and clarity. In what follows the author's responses to the reviewer’ comments are given on a point-to-point basis.
1. In the introduction part more literature review is needed.
Please note that the introduction section has been revised, and the number of references has substantially increased in the new version of the manuscript.
2. Could you explain what is the speed and positioning of x-y coordinate system?
The reviewer is right. We have tried to summarize in the revised document this point, to clarify the issue for future readers, who are referred to [Ref1].
[Ref1] Bermúdez, Mario; Martín, Cristina; González, Ignacio; Duran, Mario J.; Arahal, Manuel R.; Barrero, Federico, “Predictive current control in electrical drives: an illustrated review with case examples using a five-phase induction motor drive with distributed windings,” IET Electric Power Applications. 2020. Vol. 14. Iss. 8. Pp. 1291-1310. DOI: 10.1049/iet-epa.2019.0667
A more extensive description can be found here:
Note that any multiphase machine can be described as a set of differential equations in phase variables (currents, voltages and fluxes) using the general theory of electric machines. However, the generalization from Fortescue and Clarke laid the foundations for different mathematical transformations. Their main objective is the replacement of the original phase-variable model by equivalent equations using a reduced set of new (fictitious) variables, thus permitting the simplification of the machine model. These transformations are collected in what is usually named vector space decomposition approach, where matrix representation is conventionally adopted. For a five-phase IM, induction machine with distributed windings that are electrically displaced by 2π/5, start connected and isolated neutral point and since the rotor is squirrel-cage type, it can be treated as five windings equally displaced around the rotor circumference.
Thus, it is possible to represent the five-phase IM model in a stationary reference frame formed by a new set of five fictitious variables using the Clarke transformation. They are grouped into two two-dimensional orthogonal planes, named α-β and x-y, whose components are also orthogonal between them. The main characteristic of this new reference frame is that the different planes are completely decoupled due to the orthogonality, resulting in a simpler model that is more suitable for control purposes. In addition, the transformation of the phase variables into the new stationary reference frame provides better insight into the physical phenomena involved in the energy conversion process: the α-β plane (that includes the fundamental frequency and harmonic components of the order 10k±1, with k =1;2;3;...;∞) becomes responsible for the torque production, while the x-y plane (that includes harmonic components of the order 10k±3, with k =1;2;3;...;∞) and the z-axis (that includes harmonic components of the order 5k, with k =1;2;3;...;∞) are only related to harmonic components and stator losses in the machine.
3. The matrices are usually put in square brackets.
Solved.
4. All abbreviations should be explained (e.g. ASF, HCSMPC).
The abbreviations have been checked to avoid use of undefined ones.
5. Where are two tunings of the STD-FSMPC presented (mentioned in rows 274-280)?
The paper has been edited so that the two tunings are now introduced before their use in the text.
6. Even though this article brings interesting simulation and experimental results that support the idea of better smoothing of current, and thus torque ripples in five-phase IM achieved with Sch-FSMPC, the results should be further analyzed and discussed in section 5.
New experimental results have been added, and the discussion section has been revised in the new version of the manuscript.

Round 2
Reviewer 1 Report
Thank you for the revised manuscript.
Reviewer 3 Report
I have no further comments nor suggestions.